# Photo-induced trifunctionalization of bromostyrenes via remote radical migration reactions of tetracoordinate boron species

Chaokun Li[1], Shangteng Liao[1], Shanglin Chen[1], Nan Chen[1], Feng Zhang[1], Kai Yang[1] & Qiuling Song [1,2,3 ✉]

Tetracoordinate boron species have emerged as radical precursors via deboronation by photo-induced single electron transfer (SET) pathway. These reactions usually produce an alkyl radical and boron-bound species, and the valuable boron species are always discarded as a by-product. Given the importance of boron species, it will be very attractive if the two parts could be incorporated into the eventual products. Herein we report a photo-catalyzed strategy in which in situ generated tetracoordinated boron species decomposed into both alkyl radicals and boron species under visible light irradiation, due to the pre-installation of a vinyl group on the aromatic ring, the newly generated alkyl radical attacks the vinyl group while leaving the boron species on *ipso*-position, then both radical part and boron moiety are safely incorporated into the final product. Tertiary borons, secondary borons, *gem*-diborons as well as 1,2-diborons, and versatile electrophiles are all well tolerated under this transformation, of note, *ortho*-, *meta*- and *para*-bromostyrenes all demonstrated good capabilities. The reaction portraits high atom economy, broad substrate scope, and diversified valuable products with tertiary or quaternary carbon center generated, with diborons as substrates, C$sp^2$-B and C$sp^3$-B are established simultaneously, which are precious synthetic building blocks in chemical synthesis.

[1] Key Laboratory of Molecule Synthesis and Function Discovery, Fujian Province University, College of Chemistry and College of Materials Science at Fuzhou University, Fuzhou, Fujian 350108, China. [2] Institute of Next Generation Matter Transformation, College of Material Sciences Engineering, Huaqiao University, Xiamen, Fujian 361021, China. [3] School of Chemistry and Chemical Engineering, Henan Normal University, Xinxiang, Henan 453007, China. ✉email: qsong@hqu.edu.cn

Tetracoordinate boron species as the key intermediates have been widely existed in versatile boron-involved transformations, specifically, 1,2-metallate migration reactions and transmetallations are the most prevalent and attractive ones which usually are engaged in a nucleophilic migration[1–13]. Very recently, the alkyl tetracoordinate boron species have been used as radical precursors to participate transition-metal catalyzed cross-couplings[14–16], radical-polar cross-over reactions[17–24], Giese-type radical additions[25–33] and others[34,35]. Surprisingly, the afore-mentioned reactions inevitably have a common problem: after generation of an alkyl radical and boron-bound species under photo-induced conditions, the boron species is removed and discarded as a waste (Fig. 1A), which leads to a low atom economy. Given the importance of boron species, it will be very attractive if the two moieties could be incorporated into the eventual products, which, obviously, will be a boron-economical and sustainable process. But how to solve the problem? We envisage that if a radical acceptor could be introduced on the aromatic ring, once the alkyl radical is generated from the phenyl substituted tetracoordinate boron intermediate by the cleavage of Csp³-B bond upon irradiation, it will immediately be trapped by the radical acceptor, while leaving the boron moiety unscathed. With this hypothesis in mind, o-bromostyrene catches our eyes. At first glance, it is an electrophilic reagent, however, under the

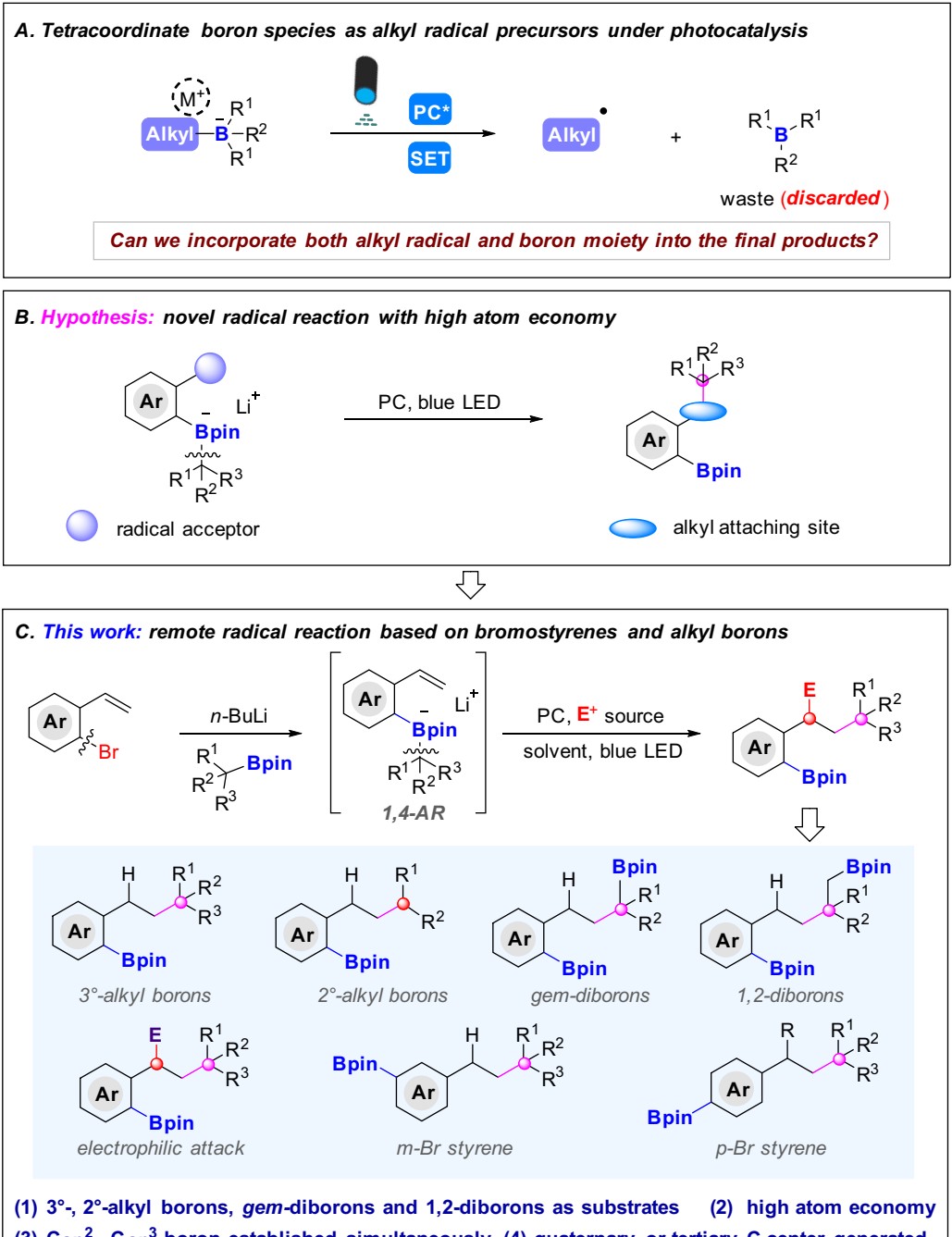

**Fig. 1 Alkyl radical precursors based on tetracoordinate boron species and radical migration reactions. A** Tetracoordinate boron species as alkyl radical precursors under photocatalysis. **B** Hypothesis: remote radical migration reaction with high atom economy. **C** Remote radical migration reaction based on bromostyrenes and alkyl borons (this work). Bpin = boronic acid pinocol ester.

**Table 1 Optimization of the reaction conditions[a].**

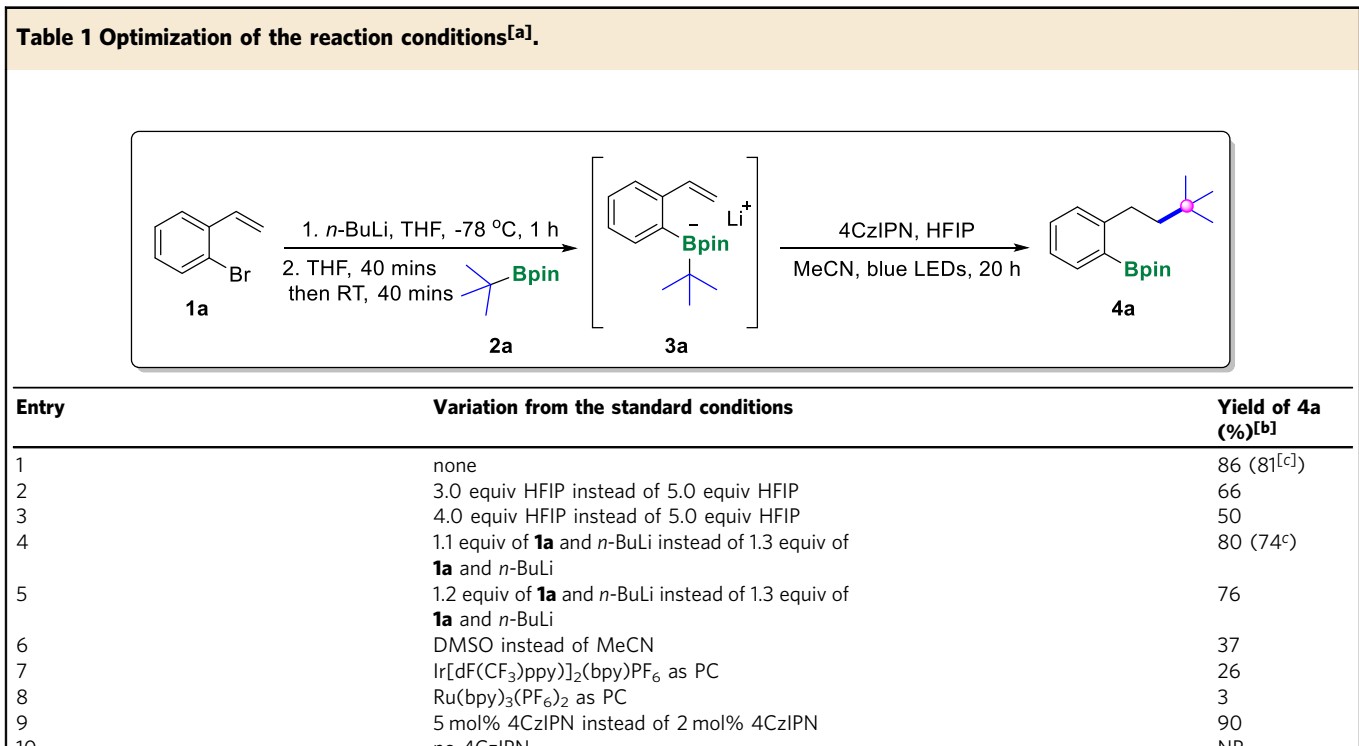

| Entry | Variation from the standard conditions | Yield of 4a (%)[b] |
|---|---|---|
| 1 | none | 86 (81[c]) |
| 2 | 3.0 equiv HFIP instead of 5.0 equiv HFIP | 66 |
| 3 | 4.0 equiv HFIP instead of 5.0 equiv HFIP | 50 |
| 4 | 1.1 equiv of **1a** and *n*-BuLi instead of 1.3 equiv of **1a** and *n*-BuLi | 80 (74c) |
| 5 | 1.2 equiv of **1a** and *n*-BuLi instead of 1.3 equiv of **1a** and *n*-BuLi | 76 |
| 6 | DMSO instead of MeCN | 37 |
| 7 | Ir[dF(CF$_3$)ppy)]$_2$(bpy)PF$_6$ as PC | 26 |
| 8 | Ru(bpy)$_3$(PF$_6$)$_2$ as PC | 3 |
| 9 | 5 mol% 4CzIPN instead of 2 mol% 4CzIPN | 90 |
| 10 | no 4CzIPN | NR |
| 11 | no blue LEDs | NR |

action of the *n*-butyl lithium reagent[36–41], the halogen-lithium exchange will convert it as a nucleophilic reagent to trap boron compounds and lead to the tetracoordinate boron species, which can perfectly fit the goal we pursue to achieve. Not surprisingly, the in situ generated tetracoordinated boron species decomposed into both alkyl radicals and boron species under visible light irradiation, due to the pre-installation of a vinyl group on the *ortho* position, the newly generated alkyl radical attacks the vinyl group while leaving the boron species on *ipso*-position, by doing these, both radical part and boron moiety were safely incorporated into the final products (Fig. 1B).

Herein, we disclose an interesting remote radical migration reaction by using the bromostyrene skeletons under the mediation of lithium reagents, all tertiary and secondary alkyl boronic esters, 1,2-diborons as well as *gem*-diborons and versatile electrophiles are well tolerated under this transformation, most remarkably, besides *ortho*-bromostyrenes, *meta*- and *para*-bromostyrenes, especially 1,1-disubstituted *para*-bromostyrenes are also suitable substrates to this transformation, thus rendering *meta*- or *para*-substituted arylboronates with new 3° or 2° carbon center generated. The reaction features high atom economy with triple functionalizations of bromostyrenes, readily accessible starting materials, broad substrate scope and diversified valuable products with tertiary or quaternary carbon center generated, gratifyingly, with 1,2-diborons and 1,1-diborons as substrates, both Csp$^2$-B and Csp$^3$-B bonds are established simultaneously in one-pot reaction, which are precious synthetic building blocks in chemical synthesis and have been demonstrated in versatile structural elaborations in our strategy (Fig. 1C).

## Results and discussions
**Investigation of reaction conditions**. We began our investigation by studying the reaction of *tert*-butyl boronic acid pinacol ester **1a** with *o*-bromostyrene (**2a**), which was under the action of the *n*-butyl lithium reagent, to form a tetracoordinate boron species **3a**.

After that, a solution of HFIP and photocatalyst 4CzIPN was added to the vessel. The reaction was conducted under blue light irradiation at room temperature for 20 h and led to the corresponding product **4a** (Table 1). After substantial optimizations for this three-step reaction, the desired product **4a** was obtained in 81% isolated yield (entry 1). Using 3.0 and 4.0 equiv of HFIP decreased the yields to 66% and 50%, respectively (entries 2 and 3). It was found that when 1.1 or 1.2 equiv of **1a** with *n*-BuLi was added, **4a** were obtained in 80% and 76% yields accordingly (entries 4 and 5). Switching the solvent from MeCN to DMSO, the yield of this reaction decreased dramatically (entry 6). When the photocatalyst was changed into Ir[dF(CF$_3$)ppy)]$_2$(bpy)PF$_6$ or Ru(bpy)$_3$(PF6)$_2$, the yields decreased to 26% and 3% correspondingly (entries 7 and 8). When the loading of the photocatalyst was increased to 5 mol%, a slight increase in this system was observed by giving **4a** in 90% yield (entry 9). However, no reaction occurred when the transformation had no photocatalyst or was performed in the darkness (entries 10 and 11). (See Supplementary Tables 1–5 in Supplementary Information for details).

To a solution of *o*-bromostyrene **1a** (0.26 mmol, 1.3 equiv) in THF (0.6 mL) was added *n*-BuLi (0.26 mmol, 1.3 equiv) at −78 °C under argon, the resulting mixture was stirred at −78 °C for 1 h, then **2a** (0.2 mmol, 1.0 equiv) in THF (0.2 ml) was added and stirred at −78 °C for 40 min and warm to room temperature for 40 min, then followed by HFIP (1.0 mmol, 5.0 equiv) and 4CzIPN (2 mol%) in MeCN (2 mL, 0.1 M). The resulting mixture was irradiated by blue LEDs for 20 h; [b] Determined by GC analysis by using dodecane as the internal standard; [c] Isolated yield.

**Synthetic scope**. With the optimal reaction conditions in hand, we proceeded to explore the generality of this strategy. Firstly, tertiary alkyl boronic esters **2** as well as *o*-bromostyrenes **1** were investigated (Fig. 2). To our delight, versatile tertiary alkyl mono-boronic esters were aptly transformed into the corresponding target products in moderate to excellent yields. As shown in

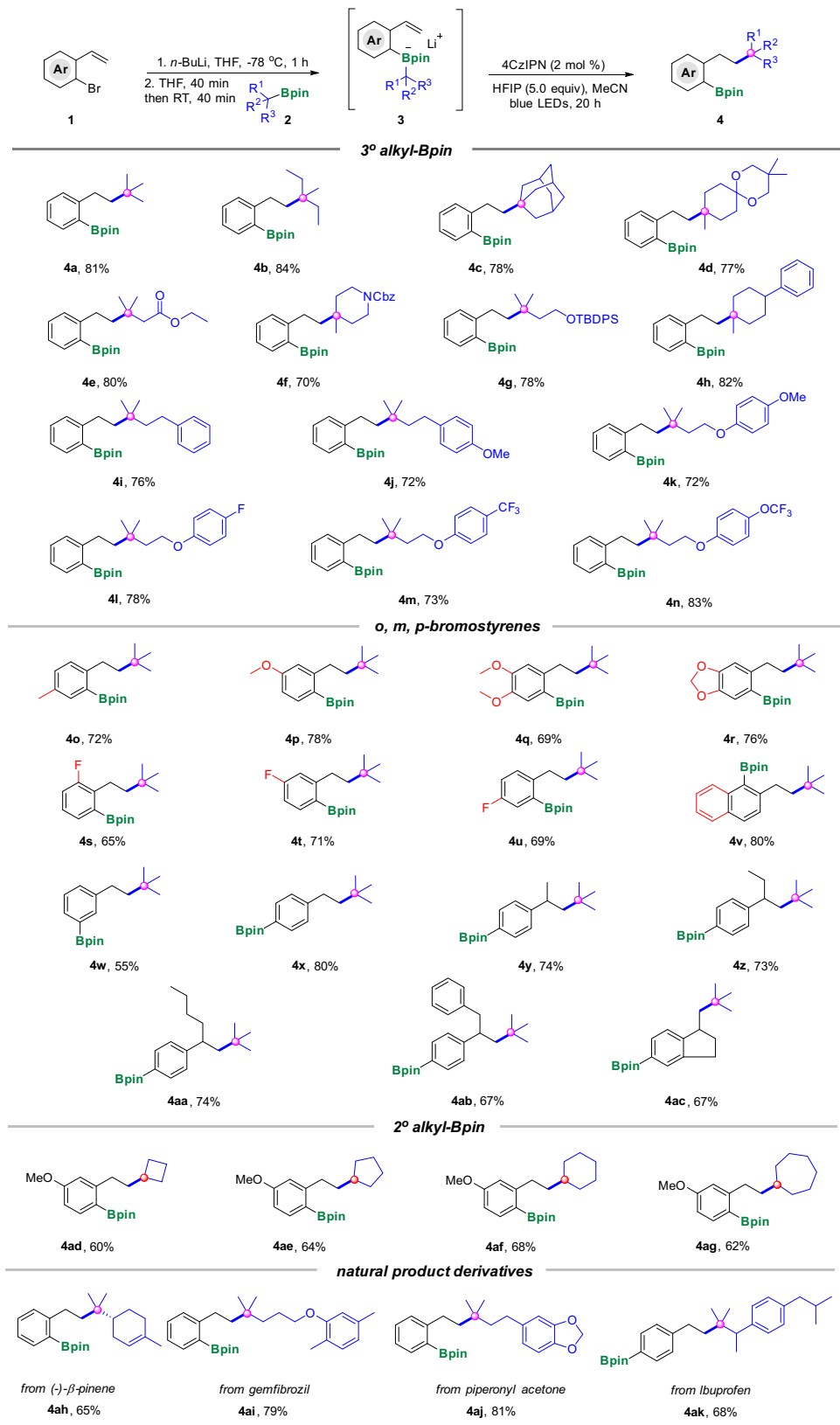

**Fig. 2 Scope of alkyl boronic esters and bromostyrenes.** [a] Reaction conditions: To a solution of bromostyrene **1** (0.26 mmol, 1.3 equiv) in THF (0.6 mL) was added *n*-BuLi (0.26 mmol, 1.3 equiv) −78 °C under argon, the resulting mixture was stirred at −78 °C for 1 h, then **2** (0.2 mmol, 1.0 equiv) in THF (0.2 ml) was added and stirred at −78 °C for 40 min and warm to room temperature for 40 min, then followed by HFIP (1.0 mmol, 5.0 equiv) and 4CzIPN (2 mol%) in MeCN (2 mL, 0.1 M). The resulting mixture was irradiated by blue LEDs for 20 h. Cbz benzyloxycarbonyl, TBDPS *tert*-butyldiphenylsilyl.

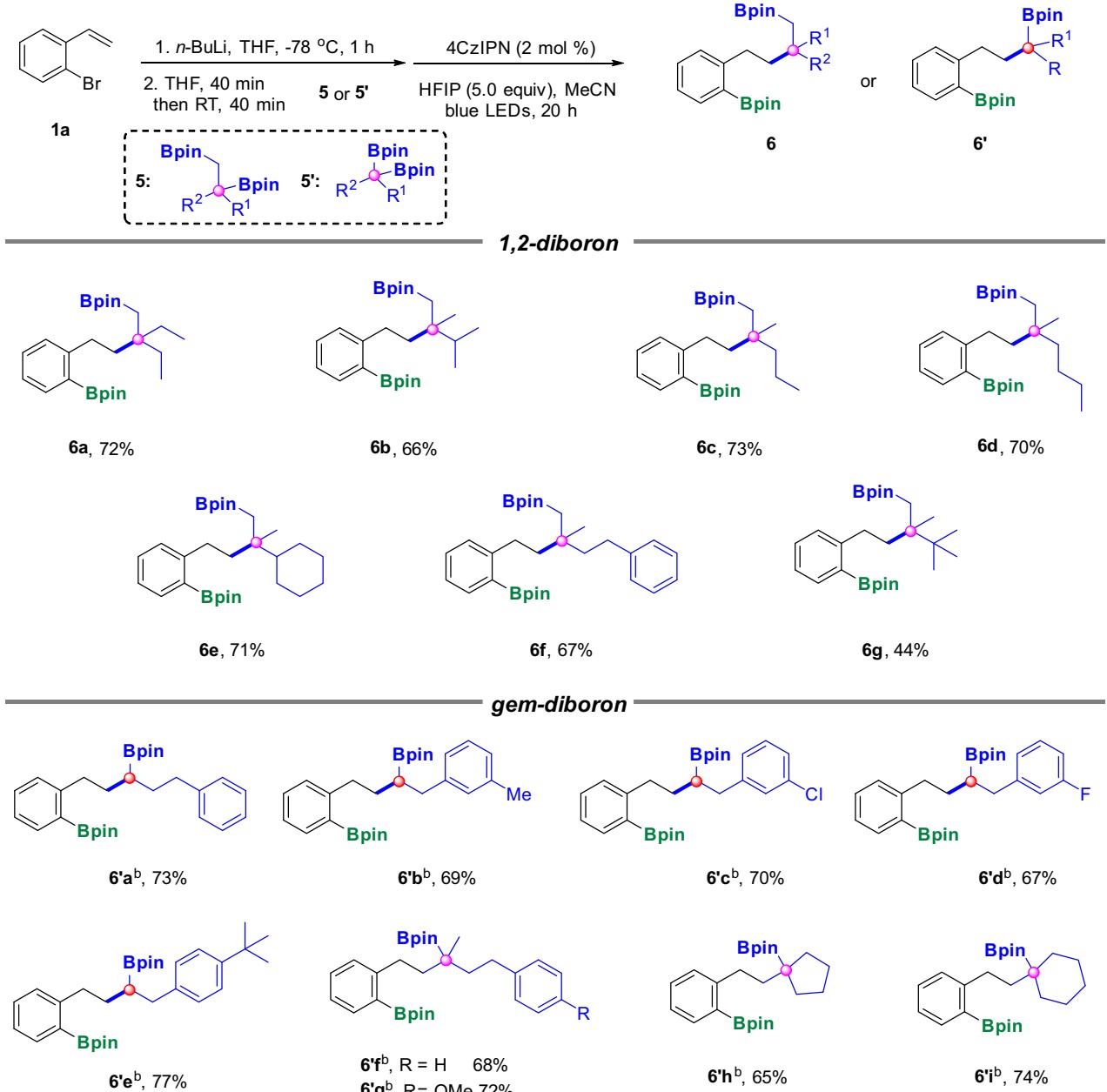

**Fig. 3 Scope of 1,2-diborons and *gem*-diborons.** [a] Reaction conditions: To a solution of *o*-bromostyrene **1a** (0.26 mmol, 1.3 equiv) in THF (0.6 mL) was added *n*-BuLi (0.26 mmol, 1.3 equiv) −78 °C under argon, the resulting mixture was stirred at −78 °C for 1 h, then **5** (0.2 mmol, 1.0 equiv) in THF (0.2 ml) was added and stirred at −78 °C for 40 min and warm to room temperature for 40 min, then followed by HFIP (1.0 mmol, 5.0 equiv) and 4CzIPN (2 mol%) in MeCN (2 mL, 0.1 M). The resulting mixture was irradiated by blue LEDs for 20 h; [b] With removing the THF at the third step.

Fig. 2, different types of tertiary alkyl mono-boronic esters were good substrates, and both the boron moiety and the tertiary aliphatic groups were smoothly incorporated into the eventual products (**4a–4n**), remarkably, along with the formation of an all-carbon quaternary carbon center.

Of note, the preparation of all-carbon quaternary carbon center are always big challenges in organic synthesis[42], with our strategy, the quaternary carbon center could be built up conveniently. For instance, besides the simple tertiary aliphatic groups, such as *tert*-butyl (**4a**), 3-methylpentyl (**4b**) and 1-adamantyl (**4c**), those ones bearing functional groups, like acetal (**4d**) and ester (**4e**), were also compatible in this transformation. Different protected groups, like Cbz-protected piperidinyl (**4f**), TBDPS-protected alcohol (**4g**) and various phenyl protected alcohols (**4k–4n**) were

also tolerant in this reaction. In addition, the substrates containing a benzene ring (**4h** and **4i**), and a methoxy group (**4j**) on the benzene ring were also good candidates for this transformation.

Subsequently, the scope of *o*-bromostyrenes was evaluated (Fig. 2). Various substituents on the *o*-bromostyrenes, such as methyl (**4o**), methoxy (**4p**) and dimethoxy (**4q–4r**) were all compatible in this reaction. Fluorine substituted substrates were also well-behaved, *ortho*-, *para*- and *meta*-substituted ones all gave good yields of the desired products (**4s–4u**). Gratifyingly, the substrate contained naphthalene ring also demonstrated good reactivity (**4v**). Most remarkably, *meta*- and *para*-bromostyrenes were also applicable to the standard conditions and the corresponding desired *meta*- and *para*-substituted boronates were obtained in decent yields (**4w–4ac**). Besides

mono-substituted alkenes, 1,1-disubstituted *para*-bromostyrenes (**4y–4ac**) were also tolerable to the standard conditions rendering the target products in satisfactory yields, which looked like the difunctionalization of vinyl group on simple monosubstituted styrenes.

Next, secondary alkyl mono-boronic esters were also screened for the transformation. Cyclic secondary boronic esters, like cyclobutanyl, cyclopentanyl, cyclohexyl and cycloheptyl were all well compatible and afforded the desired products in moderate to good yields (**4ad–4ag**). To further demonstrate the utility of this photo-induced formal radical shift reaction of boronic esters, several natural products and food additives derived alkyl boronic esters were also applied to this protocol. To our delight, these complicated structures were all tolerable under the standard conditions, rendering the corresponding desired products in good to excellent yields (**4ah–4ak**).

Organoboron compounds which contain multiple different types of C-B bonds (such as *sp/sp²/sp³* C-B bonds) in the same structure are especially remarkable molecules[43], since chemoselective and diverse C-C or C-heteroatom bonds could be sequentially built-up due to the possibility of stepwise multi-elaborations of the different types of C-B bonds, however, the efficient methods for the construction of organoboron compounds which bearing both C*sp²*-B and C*sp³*-B bonds on the same molecular scaffolds are very rare[44]. Therefore, a simple and efficient synthetic method to realize this goal becomes essential. Gratifyingly, with our current strategy, when 1,2-diborons and *gem*-diborons were employed as the substrates, the reaction worked perfectly to lead to the corresponding new diboron products containing both C*sp²*-B and C*sp³*-B bonds (Fig. 3).

Various aliphatic 1,2-diboron were tolerated well in our reaction to obtain the corresponding 1,6-diborons with one C*sp²*-B bond established together with one C*sp³*-B bond intact (**6a–6g**). And when the benzene ring was introduced on the substrate, the desired product **6f** was procured in a moderate yield. The yield of **6g** decreased significantly, possibly due to its steric hindrance from *tert*-butyl moiety, which is not conducive to the formation of intermediate tetracoordinate boron species. It is worth mentioning that the site selectivity of this reaction is very good. Only the selective reaction at tertiary position was proceeded. There are two possible pathways for this transformation: (1) tetracoordinate boron species was directly formed at tertiary boron, thus a tertiary alkyl radical was generated in a straightforward way under photo irradiation, subsequent formal 1,5-radical migration rendered an all-carbon quaternary center; (2) Aggarwal's previous report indicated that 1,2-diboronates could undergo 1,2-boron shifts of β-boryl radicals under photoredox catalysis to achieve deboronative functionalization at a more steric hindered position[32], therefore in our cases, tetracoordinate boron species might be formed at the primary boron site first, after the generation of a primary alkyl radical, a 1,2-boron shift might take place to lead to a more stable tertiary carbon radical, which might undergo subsequent radical addition to the olefin to procure the desired products.

In addition to 1,2-diborons, *gem*-diborons are also widely acces-sible boron-bearing starting materials which are compatible to the standard conditions as well to render to the desired 1,5-diborons with one C*sp²*-B bond and one C*sp³*-B bond (**6'a–6'i**). Compared with 1,2-diborons, *gem*-diborons could lead to new diborons with one C*sp²*-B bond constructed on benzene ring and

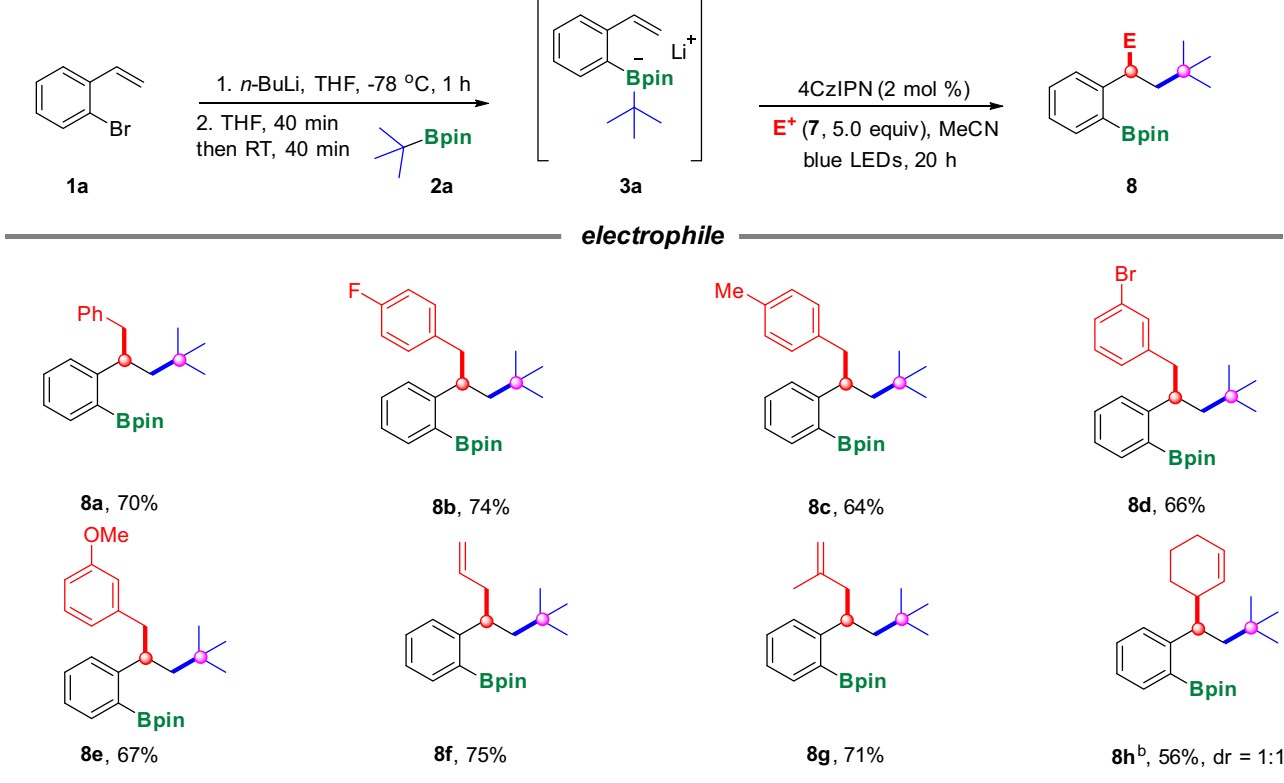

**Fig. 4 Scope of electrophiles.** [a] Reaction conditions: To a solution of *o*-bromostyrene **1a** (0.26 mmol, 1.3 equiv) in THF (0.6 mL) was added *n*-BuLi (0.26 mmol, 1.3 equiv) −78 °C under argon, the resulting mixture was stirred at −78 °C for 1 h, then **2a** (0.2 mmol, 1.0 equiv) in THF (0.2 mL) was added and stirred at −78 °C for 40 min and warm to room temperature for 40 min; then electrophile **7** (1.0 mmol, 5.0 equiv) and 4CzIPN (2 mol%) were added in MeCN (2 mL, 0.1 M) and irradiated by blue LEDs under an argon atmosphere for 20 h; [b] The dr was determined by ¹H NMR analysis.

one secondary or tertiary C$sp^3$-B bonds on the aliphatic scaffold, while 1,2-diborons procure new diborons with a primary C$sp^3$-B bond on their structure along with an all-carbon quaternary carbon center generated in situ, which prove to be a challenge in organic synthesis[43].

Of note, in the above examples, the in situ formed benzylic radicals should be reduced to carbanions in the last step, which therefore further trapped protons to render the final products. Instead of proton, the benzylic carbanion could serve as nucleophiles to react with various electrophiles to lead to difunctionalization of the olefin moiety[45], which also highlights the potential for the synthesis of more diversified and complex products. A plethora of electrophiles, such as benzyl bromide (7a), para-fluorine and para-methyl substituted benzyl bromide (7b, 7c), meta-bromine substituted benzyl bromide (7d) as well as meta-methoxy substituted benzyl bromide (7e) were all tolerated well to give the corresponding products in moderate to good yields (8a–8e). In addition, allyl bromide (7f), 2-methylallyl bromide (7g), and 2-cyclohexenyl bromide (7h) were also amenable to this system to procure the targeted products in good yields (8f–8h), which significantly increase the diversity and complexity of the final products (Fig. 4).

**Synthetic application**. To demonstrate the practical utility of these transformations, two ten times scale-up reactions were performed, furnishing the corresponding mono-boron product 4p in 58% and diboron product 6a in 52% yields (Fig. 5), whcih are a little lower than those in small-scale experiments. In order to showcase practical value of this strategy, then a series of

transformations on 4p and 6a were conducted. For example, the C$sp^2$-B bond in 4p were smoothly converted to alkyl groups via Suzuki-Miyaura couplings under the identified conditions, rendering the benzylative product 9 in 84% yield (Fig. 5a) and methylation product 10 in 86% yield (Fig. 5b)[46]. Metal-free coupling of 4p with 2-thiophenyl lithium reagent gave 11 in 74% yields (Fig. 5c)[37]. Then, the various follow-up transformations on 6a were also performed. First, the two boron moieties could be oxidized into bis-hydroxyl groups to lead to 1,6-diol 12 in 88% yield (Fig. 5d)[24]. Selective Suzuki-Miyaura couplings on C$sp^2$-B moiety of 6a was produced a phenylative product 13 in 92% yield while leaving the C$sp^3$-B bond intact (Fig. 5e)[46], which then underwent the carbon elongation reaction to lead to new boron 14 (Fig. 5f)[47], meanwhile, compound 13 could also experience the vinylation on C$sp^3$-Bpin species to render compound 15 (Fig. 5g)[47] (See Supplementary Figs. 8–14 in Supplementary Information for details).

**Mechanistic studies**. To thoroughly understand this transformation, we did some control experiments and mechanistic studies (Fig. 6) (See Supplementary Figs. 15–19, Supplementary Table 6 in Supplementary Information for details). When 2,2,6,6-tetramethyl-1-piperidinyloxy (TEMPO) was added to our system, the formation of 4a was significantly inhibited, and the tert-butyl radical−TEMPO adduct 16 was detected by GCMS, indicating that radicals might be involved in our transformation (Fig. 6A). To figure out the source of protons in the reaction, $d_8$-THF and $d_3$-MeCN as solvents were added to the reaction separately, however, no any deuterium-labeled products were detected, thus

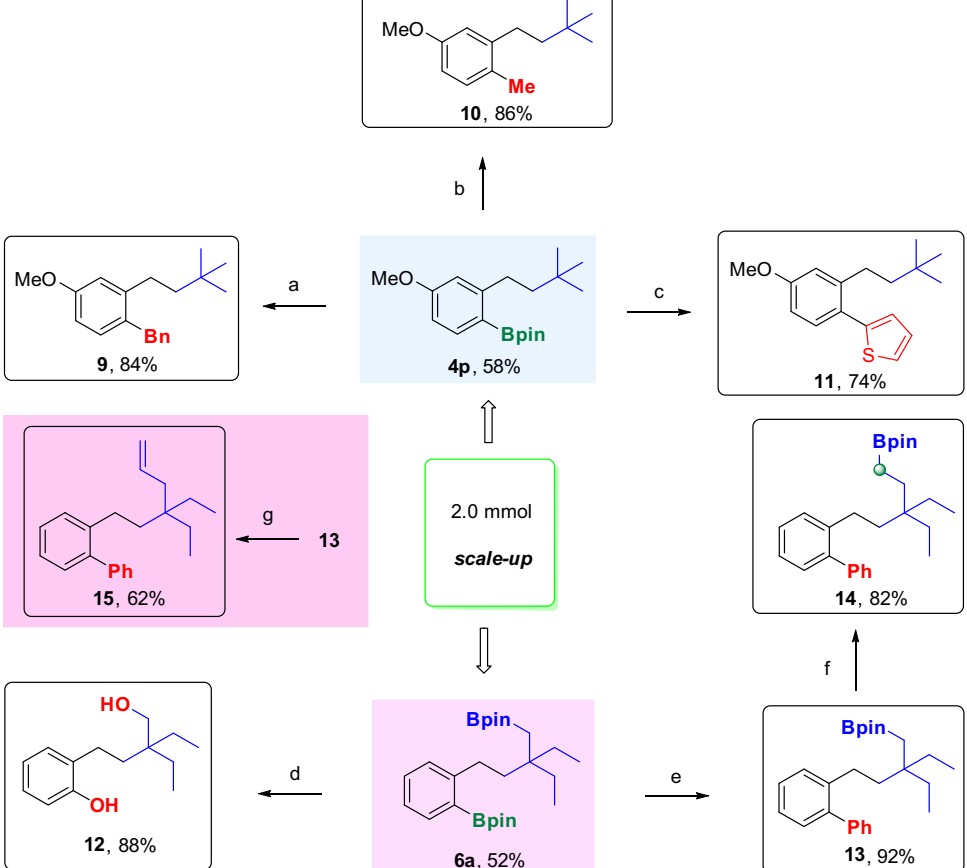

**Fig. 5 Scale-up reactions and synthetic applications. a** benzyl bromide, Pd(PPh$_3$)$_4$, Cs$_2$CO$_3$, THF/H$_2$O, 60 °C, 24 h; **b** methyl iodide, Pd$_2$(dba)$_3$, (2-tol)$_3$P, K$_2$CO$_3$, DMF/H$_2$O, 80 °C, 24 h; **c** −78 °C, n-BuLi, thiophene, then rt, 1 h; −78 °C, R-Bpin, 1 h; NBS, 1 h; **d** NaOH, H$_2$O$_2$, THF, 0 °C to rt, 4 h; **e** PhI, Pd(PPh$_3$)$_2$Cl$_2$, Cs$_2$CO$_3$, THF/H$_2$O, 80 °C, 6 h; **f** ClCH$_2$I, n-BuLi, THF, −78 °C to rt; **g** vinylmagnesium bromide, I$_2$ in MeOH, −78 °C to rt.

**A. radical capture experiment**

**1a** + **2a** → standard conditions / TEMPO (1.5 equiv) → **4a** (n.d.) + **16** (detected by GCMS)

**B. deuteration experiment**

**1a** + **2a** → standard conditions → **4a-D**

| Deuterium source | Deuteration rate |
|---|---|
| $d_8$-THF | 0 |
| $d_3$-MeCN | 0 |
| $d_2$-HFIP | 72% |

**C. radical-clock experiment**

**1a** + from (-)-β-pinene **2ah** → standard conditions → **4ah**, 65%

**D. validation experiment**

**17** → t-BuLi / THF → 4CzIPN, HFIP / MeCN, blue LEDs → **4a**, 82 %

**E. cross-over experiment**

**1a** + **2a** → standard conditions / **18** 1.0 equiv → **4a** + **19** (3 : 1)

**1a** + **2a** → standard conditions / **1a** 1.0 equiv → **4a** + **20** (1 : 1.3)

ratio detected by GCMS

**Fig. 6 Control experiments. A** Radical capture experiment; (**B**) Deuteration experiment; (**C**) Radical-clock experiment; (**D**) Validation experiment; (**E**) Cross-over experiment.

the possibility of solvents as proton sources could be ruled out. When $d_2$-HFIP was added in the reaction mixture, the corresponding product **4a–D** was obtained with 72% deuterium incorporation at the benzylic position (Fig. 6B), which suggested that HFIP should be the source of proton in our transformation. We subsequently turned our attention to the radical-clock experiments (Fig. 6C). With (−)-β-pinene derived intramolecular homolytic ring-opening pathway was involved in our transformation, thus further confirm the radical process. To

further validate the formation of tetracoordinate boron species as mono-boronate **2ah** as the substrate, a ring-opening product **4ah** was obtained smoothly[32]. The experiment suggested that an the key intermediate, we synthesized *ortho*-vinylphenyl boronic ester **17** and subjected it to our standard conditions, to our delight, the corresponding product **4a** was obtained in 82% yield (Fig. 6D), which clearly demonstrated that tetracoordinate boron species was the key intermediate for our strategy. Cross-over experiments were further carried out to figure out the reaction patterns

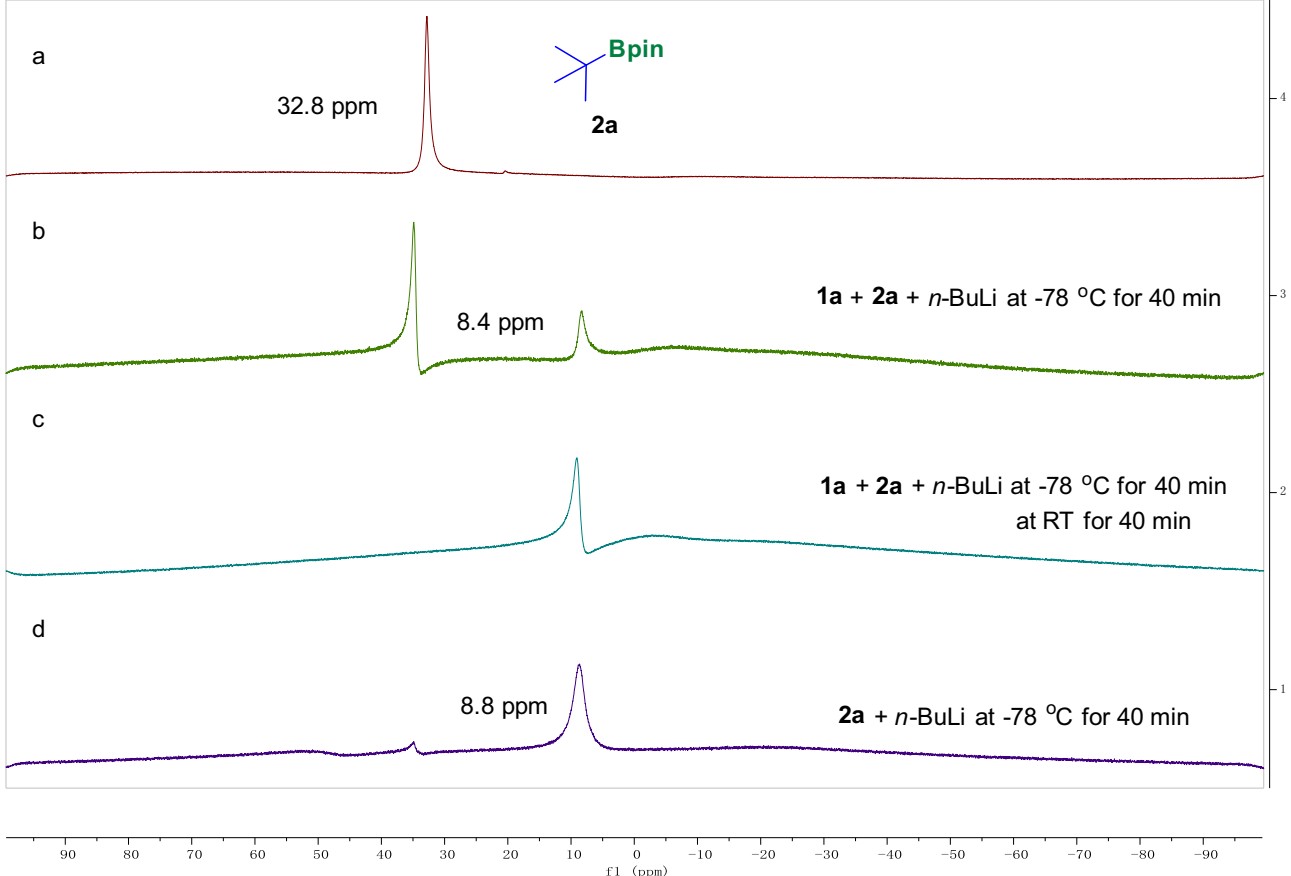

**Fig. 7 [11]B NMR analysis of the reaction in _d_8-THF. a** The boron peak of _tert_-butylboronic acid pinocol ester (_tert_-BuBpin, **2a**); **b 1a** and _n_-BuLi were added to the solution of **2a** and stirred at −78 °C for 40 min; **c** After (**b**) the reaction was allowed to warm to room temperature and stir for another 40 min; (**d**) Only added _n_-BuLi to the solution of **2a** and allowed the mixture was stirred at −78 °C for 40 min.

(Fig. 6E). In the first case, styrene (**18**) was added to the standard system and it turned out that in addition to the formation of desired product **4a**, compound **19** which was generated from the _tert_-butyl radical attacking on the vinyl group of styrene was also obtained and the ratio between compound **4a** and **19** was ca. 3:1; in the second case, instead of adding styrene (**18**), the starting material **1a** was added to the standard system and once again the corresponding _tert_-butyl radical trapping product **20** was detected (Fig. 6E), these two cross-over experiments indicated that the method should undergo an intermolecular pathway. We also did the luminescence quenching experiment and cyclic voltammetry measurement to improve the relevant data (See Supplementary Figs. 20–21 in Supplementary Information for details).

To gain further insights into the mechanism and validate the key intermediate of this transformation, we carried out several experiments and detected them by [11]B NMR analysis (Fig. 7)[24,48]. The boron peak of _tert_-butylboronic acid pinocol ester (_tert_-BuBpin, **2a**) was at 32.8 ppm on [11]B NMR (Fig. 7a). When **1a** and _n_-BuLi were added to the solution of **2a** and stirred at −78 °C for 40 min, a new peak at 8.4 ppm was observed, which was assigned as a tetracoordinate boron species (Fig. 7b). Then the reaction was allowed to warm to room temperature and stir for another 40 min, the peak at 32.8 ppm completely disappeared and only the peak at 8.4 ppm remained (Fig. 7c). Moreover, when we only added _n_-BuLi to the solution of **2a** and allowed the mixture was stirred at −78 °C for 40 min, an obvious signal at 8.8 ppm showed up (Fig. 7d), which was the tetracoordinate boron intermediate generated from _t_-BuBpin and _n_-BuLi, yet was a little bit different from the tetracoordinate boron intermediate in our standard

reaction mixture. Therefore, it could clearly prove that a new tetracoordinate boron intermediate was formed in our system, thus validate our hypothesis.

**Proposed mechanism.** Based on these control experiments and previous investigations[49], we proposed a plausible reaction mechanism, which may undertook the route of path A or path B (Fig. 8). Firstly, under the action of _n_-BuLi, _o_-bromostyrene **1** generates new lithium salt **I**. Then alkyl boronic esters **2** is added and tetracoordinate boron intermediate **3** is formed in situ. Subsequently, single-electron transfer (SET) between the excited state photocatalyst 4CzIPN* ($E_{1/2}$ [PC*/P·−] = 1.35 V vs. SCE in MeCN[49]) and tetracoordinate boron intermediates renders an alkyl radical **II** and the anionic catalyst radical. Then, the intermolecular radical addition to the olefinic moiety of _o_-bromostyrene (**III**) leads to intermediate **IV**. In the path A, next, intermediate **IV** undergoes SET reduction with the reduced photocatalyst (P·−, $E_{1/2}$ [PC/P·−] = −1.21 V vs. SCE in MeCN for 4CzIPN) to give the benzylic carbanion **V** and the photocatalyst 4CzIPN, the added electrophile reacts with the carbanion **V** to procure the final product **VI**. In the path B, aryl bromide undergoes SET reduction with the reduced photocatalyst to give the intermediate **VII** and the photocatalyst 4CzIPN. The intermediate **VII** dissociates a bromide anion and a benzyl radical, which combines with **IV** and undergoes a radical-radical coupling process to get the final product **VIII**.

In conclusion, we disclosed a remote radical migration reaction by using the various bromostyrenes under the mediation of lithium reagents, both tertiary and secondary boronic esters as

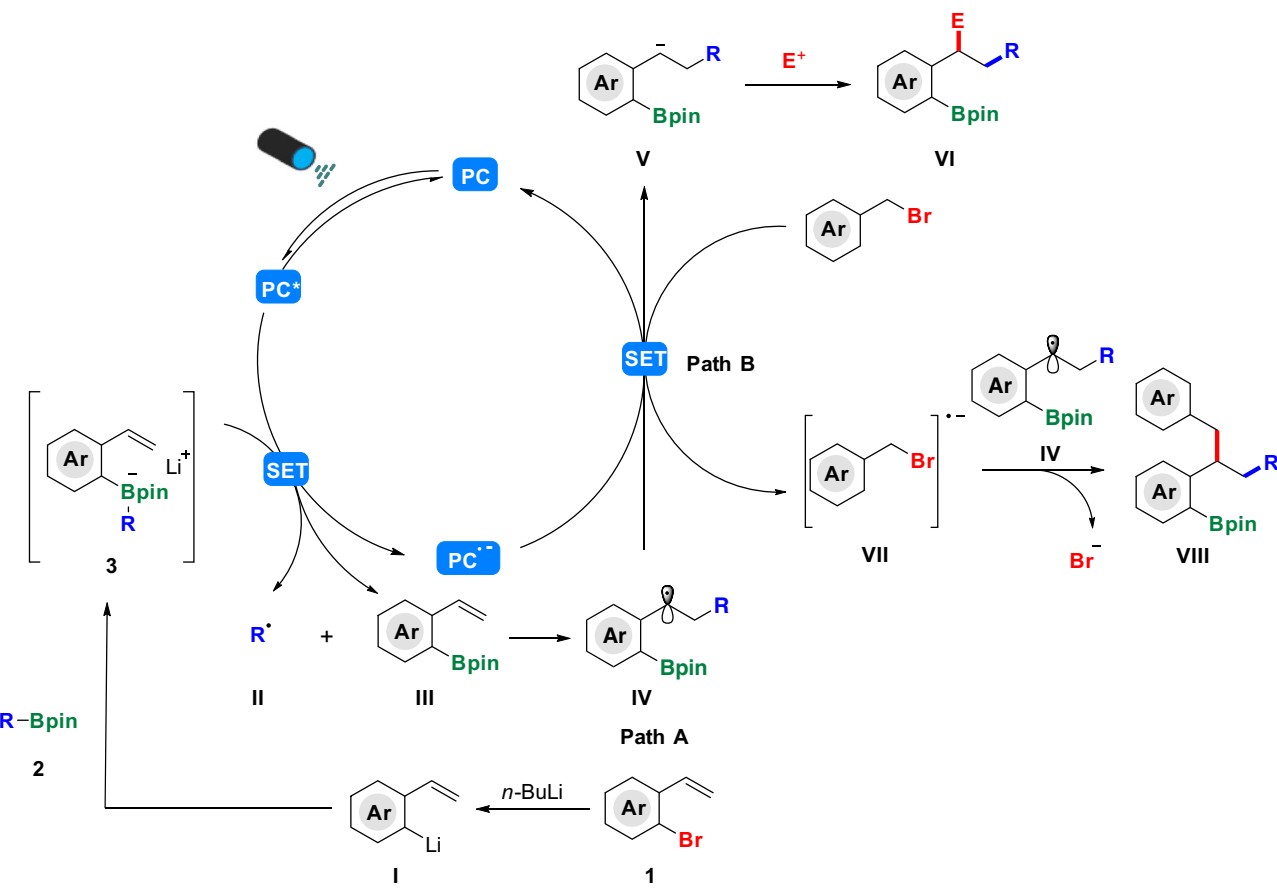

**Fig. 8 Proposed mechanism.** Ar = aromatic ring; R = alkyl chain; SET = single electron transfer; Bpin = boronic acid pinocol ester; PC = photocatalyst; PC* = the excited state photocatalyst; P·⁻ = the reduced state of the photocatalyst; E = electrophile.

well as 1,2-diborons and *gem*-diborons were well tolerated under this condition, meanwhile, versatile electrophiles could also be employed in our method to lead to difunctionalization of vinyl groups. Of note, this protocol solved a persistent issue in such tetracoordinate boron species-involved radical reaction and both the alkyl part and boron moiety in the starting organoboron compounds are smoothly incorporated into the eventual products. Meanwhile, this protocol could readily construct all-carbon quaternary carbon center as well as render new type of diborons bearing both $Csp^3$-B and $Csp^2$-B bonds which could be transformed into versatile functionalities chemoselectively. Most remarkably, the tertiary $Csp^3$-B bond, which is a significant challenge in boron chemistry could be readily accessible in our method as well. Our method represents the first remote radical migration reaction by the in situ generation of tetracoordinate boron intermediate and versatile valuable products could be constructed efficiently. This strategy portraits high atom economy, broad substrate scope, and diversified products with tertiary or quaternary carbon center generated, both $Csp^3$-B and $Csp^2$-B bonds constructed in one-pot strategy, we think this work will inspire more boron-economical reaction based on tetra-coordinate boron intermediate and design more innovative transformations which will be conducive to chemical synthesis, pharmaceuticals as well as material sciences.

## Methods

**General procedure for synthesis of aryl boronic acid pinocol esters from tertiary or secondary alkyl mono-boronic esters and different electrophiles.**
To a flame-dried Schlenk tube were added bromostyrene (0.26 mmol, 1.3 equiv) and THF (0.6 mL) and the resulting solution was cooled to −78 °C. Subsequently,

*n*-BuLi (0.26 mmol, 1.3 equiv) was added dropwise and the resulting mixture was stirred at −78 °C for 1 h, before dropwise addition of tertiary or secondary alkyl mono-boronic esters (0.2 mmol, 1.0 equiv, in 0.2 mL THF) to the solution of aryllithium reagent. And the resulting mixture was stirred at −78 °C for 40 min, then allowed it to ambient temperature and stirred for another 40 min. Next, without removing the THF, the 4CzIPN (2 mol%), HFIP (or other electrophiles, 1.0 mmol, 5.0 equiv), MeCN (2 mL) was added to Schlenk tube under argon, after which the Schlenk tube was sealed with parafilm and the mixture was stirred vigorously under blue LED irradiation for 20 h. The reaction mixture was diluted with EtOAC and the solution washed with saturated aqueous NH₄Cl (with saturated aqueous NaHCO₃, when alkyl bromide as electrophile), water and brine. The combined organic layers were dried over Na₂SO₄ and concentrated under reduced pressure. The crude product was then purified by flash column chromatography.

**General procedure for synthesis of aryl boronic acid pinocol esters from 1,2-diborons or *gem*-diborons.**
To a flame-dried Schlenk tube were added *o*-bromostyrene (0.26 mmol, 1.3 equiv) and THF (0.6 mL) and the resulting solution was cooled to −78 °C. Subsequently, *n*-BuLi (0.26 mmol, 1.3 equiv) was added dropwise and the resulting mixture was stirred at −78 °C for 1 h, before dropwise addition of 1,2-diborons or *gem*-diborons (0.2 mmol, 1.0 equiv, in 0.2 mL THF), with removing the THF when *gem*-diborons as substrate, to the solution of aryllithium reagent. And the resulting mixture was stirred at −78 °C for 40 min, then allowed it to ambient temperature and stirred for another 40 min. Next, without removing the THF, the 4CzIPN (2 mol%), HFIP (1.0 mmol, 5.0 equiv), MeCN (2 mL) was added to Schlenk tube under argon, after which the Schlenk tube was sealed with parafilm and the mixture was stirred vigorously under blue LED irradiation for 20 h. The reaction mixture was diluted with EtOAC and the solution washed with saturated aqueous NH₄Cl, water and brine upon completion. The combined organic layers were dried over Na₂SO₄ and concentrated under reduced pressure. The crude product was then purified by flash column chromatography.

## Data availability
The characterization and NMR data that support the findings of this study are available within the article and its Supplementary Information files.

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

## Acknowledgements

Financial support from the National Natural Science Foundation of China (21772046, 21931013) (to Q.S.) and Open Research Fund of School of Chemistry and Chemical Engineering, Henan Normal University (to Q.S.) are gratefully acknowledged.

## Author contributions

Q.S. conceived and directed the project. C.L. designed and performed experiments. S.L., S.C., N.C., F.Z. & K.Y. helped with the collection of some new compounds and data

analysis. Q.S. & C.L. wrote the paper with input from all other authors. All authors discussed the results and commented on the manuscript.

## Competing interests
The authors declare no competing interests.
