## [Peer Review File · Nature Communications]

REVIEWER COMMENTS

Reviewer #1 (Remarks to the Author):

In the present manuscript, Q. Song et al. reported a photo-induced functionalization of bromostyrenes via radical addition to the alkene functionality/ borylation at the lithiated ortho-position. The reported chemical transformation is very interesting, since it stands for the first one incorporating both the alkyl radical and the boron species into the final product. The manuscript is well organized, and the results clearly shown. Accordingly, it could be suitable for publication in Nature Commun. after the following revisions will be addressed:

- 1) Please check the manuscript for typos. There are a lot all over the text. For example, in the abstract: "tolerable" would be better changed into "tolerated"; "were also demonstrated good capabilities" sounds not correct. In general, English should be carefully revised.
- 2) Abstract: it's quite long. It would be more impactful if shortened.
- 3) Figure 1: Csp2 and Csp3: format apex
- 4) Table 1: Under the first arrow, it would be better to place a semicolon after THF, 40 min. (or "then")
- 5) Table 1, caption: specify reaction times and temperatures after the addition of 2a as depicted in the reaction scheme
- 6) What about primary alkyl boronic esters? Are the suitable starting materials as well as secondary and tertiary ones? Did the authors investigate this issue?
- 7) Figure 2: "1" and "2" should be in bold; indicate the equivalents of HFIP in the caption; if RT, please specify
- 8) Place Figure 3 after mentioning it in the text; "gem" should be in italic; "5" in bold; specify molarity of MeCN
- 9) For the sake of clarity, it would be better to draw the diboron species in Figure 3
- 10) Figure 4: place the figure after citing it in the text
- 11) The authors described the benzyl bromides used as electrophiles and numbered them; however, it is confusing to not find any figure. If they want to keep the numbering, a Figure should be added.
- 12) When the electrophile is an alkyl bromide, the authors did not add a base. Which is the species neutralizing the HBr formed during the reaction?
- 13) Reaction mechanism with benzyl bromides: why the authors excluded a mechanism involving a radical-radical coupling? The benzyl bromide could be indeed reduced to radical anion by regenerating the photocatalyst, thus giving a benzyl radical and a bromide anion. The benzyl radical could then

quench the ortho-boron benzyl radical generated upon addition of the alkyl radical (from the boronate) to the styrene double bond.

14) Albeit 3 is an in situ formed species, would it be possible to perform Stern-Volmer quenching experiments with photocatalyst 4CzIPN? Did the authors already try?

15) Reaction mechanism: why the authors excluded that formation of the product from the benzyl radical could be formed upon H-atom abstraction from HFIP?

16) The authors stated that interception with various electrophiles is possible, however the electrophiles' scope seems to be limited to benzyl-/alkyl bromides. What about other ones such as, for example, acyl chlorides or chloroformate?

17) Blue LEDs: specify the potency also in the manuscript. In the Supporting Information file, the picture shows 2 LED lamps: if it is necessary to use 2, it is important to specify it to assure reproducibility of the synthetic procedure.

SUPPORTING INFORMATION

18) F254: 254 should be formatted as subscript

19) Specify literature references for the reported alkyl-Bpin, 1,2-diboron, and 1,1-diboron starting materials

20) Section 6.1 (Na₂SO₄, "4" should be formatted as subscript)

21) HRMS is missing for product 11; in the ¹H-NMR characterization data the integration of signal at 0.85 is indicated as "39": delete 3

22) For "hexane" sometimes is indicated "n" and sometimes not. Add "n" in italic where it is not indicated.

23) Please show the GC-MS spectrum of TEMPO-adduct 16, along with comparison between calculated and found mass value

24) 2ah: is it commercially available or the authors synthesized it?

25) Section 7.5: replace "dropwisely" with "dropwise"

26) ¹³C-NMR spectrum of 4ad seems to be oddly cut. Please provide the original FID file

27) Caption of NMR spectra: o-, m-, and p- should be formatted as italic

Reviewer #2 (Remarks to the Author):

The manuscript reported by Song et al. describes photo-induced trifunctionalization of bromostyrenes via formal remote radical migration reactions of tetracoordinate boron species. This is an interesting and useful piece of work on the radical chemistry in using tetracoordinate boron species. Especially, the generation of α -boryl carbon radical from gem-diboron is very important in synthetic organic chemistry. This paper is acceptable as an article of Nature communications after the following revisions.

- 1) The title is not correct. This reaction is obviously the radical transfer reaction. So, the author should revise the title.
- 2) If possible, the author should indicate the oxidation potential of borate anion 3 and the reduction potential of benzylic radical IV. In addition, although the excited-state redox potential of 4CzIPN is well known, the author should indicate its value in the manuscript, as it would be very useful for the readers.
- 3) In figure 8, the structure of II is not correct. The author should revise the correct structure.
- 4) In the scale-up experiments, the yields of 4p and 6a are lower than those in small-scale experiments. What are the reasons for them?
- 5) In SI, ¹¹B NMR spectra, for example 6'a, 6'b, 6'c, 6'd, 6'e, and 6'g, are strange. Please re-check those ¹¹B NMR spectra.

Reviewer #3 (Remarks to the Author):

First of all, I would say that Song and coworkers presented a nice design of photoinduced radical transformation of tetracoordinated pinacol boronates. Different from the few previous reports, the smart utilization of a vinyl group containing aryllithiums (in-situ from aryl bromides) as the "activation reagent" allows the re-assembly of the "side" arylboronates with the generated alkyl radical to achieve a formal remote migrative bond-forming process. A very broad range of alkyl boronates were well used in this transformation. The site diversity on the olefinic aryllithium was also achievable. Gem-diborons and 1,2-diborons result in various aryl-B & alkyl-B containing diboron compounds. I believe this will be a valuable organoboron chemistry in chemical synthesis. I recommend its publication in Nature Communications after addressing the following issues.

1. In Table 1, as the data are the variations from the standard conditions, please present the standard condition in the chemical equation in details. In the entries (2, 3, 4, 5), illustrating the conditions clearly about instead of what conditions of the standard.
2. In the proposed mechanism, the intermediate II was wrong, please modify it. The arrows between II and III should be a plus sign.

3. What about a none-terminal alkenyl group-containing aryllithium? Please present one example to show the possibility.

4. Many of the citations are not in proper format. Please check them carefully.

Point-by-Point Response

REVIEWER COMMENTS

Reviewer #1 (Remarks to the Author):

In the present manuscript, Q. Song et al. reported a photo-induced functionalization of bromostyrenes via radical addition to the alkene functionality/ borylation at the lithiated ortho-position. The reported chemical transformation is very interesting, it stands for the first one incorporating both the alkyl radical and the boron species the final product. The manuscript is well organized, and the results clearly shown. Accordingly, it could be suitable for publication in Nature Commun. after the revisions will be addressed:

Response: We sincerely thank this reviewer for his/her favorable comments on our submitted manuscript, we really appreciate it.

1) Please check the manuscript for typos. There are a lot all over the text. For example, in the abstract: “tolerable” would be better changed into “tolerated”; “were also demonstrated good capabilities” sounds not correct. In general, English should be carefully revised.

Response: We sincerely thank this reviewer for pointing it out, we went through the entire manuscript and made corresponding corrections. Please see our revised manuscript.

2) Abstract: it’s quite long. It would be more impactful if shortened.

Response: We thank this reviewer for the constructive comments. We have trimmed down the abstract. Please see our revised manuscript.

3) Figure 1: Csp2 and Csp3: format apex

Response: We sincerely thank this reviewer for pointing it out, we have made revision accordingly. Please see our revised manuscript.

4) Table 1: Under the first arrow, it would be better to place a semicolon after THF, 40 min. (or “then”)

Response: Sorry for the negligence, we have corrected this mistake. Please see our revised manuscript.

5) Table 1, caption: specify reaction times and temperatures after the addition of 2a as depicted in the reaction scheme

Response: Sorry for the negligence, we have corrected this mistake. Please see our revised manuscript.

6) What about primary alkyl boronic esters? Are the suitable starting materials as well

as secondary and tertiary ones? Did the authors investigate this issue?

Response: We thank this reviewer for this question. Actually, during scope exploration, we tried primary alkyl boronic esters, and no target product was detected, probably because the radical generated by the secondary and tertiary alkyl boronic esters are more stable.

7) Figure 2: “1” and “2” should be in bold; indicate the equivalents of HFIP in the caption; if RT, please specify

Response: Sorry for the negligence, we have corrected these mistakes. Please see our revised manuscript.

8) Place Figure 3 after mentioning it in the text; “gem” should be in italic; “5” in bold; specify molarity of MeCN

Response: We sincerely thank this reviewer for pointing it out, we have corrected these mistakes. Please see our revised manuscript.

9) For the sake of clarity, it would be better to draw the diboron species in Figure 3

Response: Thanks for your constructive comments, we have corrected these mistakes. Please see our revised manuscript.

10) Figure 4: place the figure after citing it in the text

Response: We sincerely thank this reviewer for pointing it out, we have corrected this mistake. Please see our revised manuscript.

11) The authors described the benzyl bromides used as electrophiles and numbered

them; however, it is confusing to not find any figure. If they want to keep the numbering, a Figure should be added.

Response: Thanks for your careful reading and constructive comments, we have corrected these mistakes and added Figure 4 at the end of the paragraph. Please see our revised manuscript.

12) When the electrophile is an alkyl bromide, the authors did not add a base. Which is the species neutralizing the HBr formed during the reaction?

Response: We thank this reviewer for pointing it out. When we used alkyl bromides as electrophiles, we added saturated sodium bicarbonate solution to neutralize the in situ formed HBr at the final quenching-step. We have updated the experimental steps. Please see our revised supporting information.

13) Reaction mechanism with benzyl bromides: why the authors excluded a mechanism involving a radical-radical coupling? The benzyl bromide could be indeed reduced to radical anion by regenerating the photocatalyst, thus giving a benzyl radical and a bromide anion. The benzyl radical could then quench the ortho-boron benzyl radical generated upon addition of the alkyl radical (from the boronate) to the styrene double bond.

Response: We deeply thank this reviewer for his/her inspiring suggestions. We have done a control experiment to verify it, but did not get a conclusive result or intermediate, so we cannot rule out our previous pathway, therefore we combined the two possible pathways and updated our proposed mechanism. We thank this reviewer again for the suggestions.

14) Albeit 3 is an in situ formed species, would it be possible to perform Stern-Volmer quenching experiments with photocatalyst 4CzIPN? Did the authors already try?

Response: We thank this reviewer for pointing it out. We have added the luminescence quenching experiments. Steady-state emission spectra were acquired using an Edinburgh Instruments, FLS 920 spectrometer. In a typical experiment, the emission spectrum of a 1×10^{-4} M solution of in THF was collected. Photocatalyst 4CzIPN (0.01 mmol, 7.9 mg) was dissolved in THF (100 mL) to set the concentration is 1×10^{-4} M. *t*-BuBpin (0.1 mmol) and (2-vinylphenyl)lithium (0.13 mmol), which was pre-prepared, were dissolved in 2 mL THF and stirred for 1 h at -78 °C, warmed to room temperature for 40 mins and diluted to 25 mL to set the concentration is 4×10^{-3} M. The Schlenk tube was charged with 2 mL 4CzIPN (1×10^{-4} M). 1000 μ L, 700 μ L, 500 μ L, 300 μ L, 200 μ L, 100 μ L, 50 μ L, 0 μ L **Complex 3** were added and the fluorescence spectra were collected, respectively.

The relative intensity I_0/I was calculated as a function of quencher concentration, where I_0 is the luminescence intensity in the absence of quencher, while I is the intensity in the presence of the quencher.

4CzIPN quenched by Complex 3

Stern-Volmer plot

15) Reaction mechanism: why the authors excluded that formation of the product from the benzyl radical could be formed upon H-atom abstraction from HFIP?

Response: We deeply thank this reviewer for his/her suggestions. We have done a control experiment. In the third step, we did not add HFIP, waited for the system to react under blue light for 20 h, and finally took it out, then added 5.0 equiv of HFIP, and finally we still got the target product with yields of 68%. In the absence of HFIP, the benzyl radical should be either self-quenched or reduced to benzyl anion during the 20 h under irradiation, it could not keep radical status. Therefore, we excluded that formation of the product from the benzyl radical which was formed upon H-atom abstraction from HFIP.

16) The authors stated that interception with various electrophiles is possible,

however the electrophiles' scope seems to be limited to benzyl-/alkyl bromides. What about other ones such as, for example, acyl chlorides or chloroformate?

Response: We thank this reviewer for the constructive comments. We have tried using MOMCl, TsCl, TBDPSCI, TMSCl and benzoyl chloride as electrophiles, no desired final products were obtained.

17) Blue LEDs: specify the potency also in the manuscript. In the Supporting Information file, the picture shows 2 LED lamps: if it is necessary to use 2, it is important to specify it to assure reproducibility of the synthetic procedure.

Response: Sorry for the negligence. The reasons for us to use 2 LED lamps was just trying to make the irradiation more uniform. Because there are usually six or eight Schlenk tubes on the reaction tube rack. We have changed these pictures. Please see our revised supporting information.

SUPPORTING INFORMATION

18) F254: 254 should be formatted as subscript

Response: We sincerely thank this reviewer for pointing it out, we have corrected this mistake. Please see our revised supporting information.

19) Specify literature references for the reported alkyl-Bpin, 1,2-diboron, and 1,1-diboron starting materials

Response: Sorry for the negligence, we have specified literature references for these starting materials. Please see our revised supporting information.

20) Section 6.1 (Na₂SO₄, “4” should be formatted as subscript)

Response: We sincerely thank this reviewer for pointing it out, we have corrected this mistake. Please see our revised supporting information.

21) HRMS is missing for product 11; in the ¹H-NMR characterization data the integration of signal at 0.85 is indicated as “39”: delete 3

Response: Sorry for the negligence. We have added the HRMS data of product **11** in the supporting information and deleted “3”. Please see our revised supporting information.

HRMS (EI-QTOF) m/z : [M]⁺ Calcd. for Chemical Formula: C₁₇H₂₂OS 274.1391; Found 274.1394.

22) For “hexane” sometimes is indicated “n” and sometimes not. Add “n” in italic where it is not indicated.

Response: Sorry for the negligence. We have corrected these mistakes. Please see our revised supporting information.

23) Please show the GC-MS spectrum of TEMPO-adduct **16**, along with comparison between calculated and found mass value

Response: Sorry for the negligence. We have added the GC-MS spectrum of TEMPO-adduct **16** in the supporting information. Chemical Formula: C₁₃H₂₇NO Exact Mass: 213; Found 213.

24) **2ah**: is it commercially available or the authors synthesized it?

Response: We sincerely thank this reviewer for pointing it out, for **2ah**, we synthesized it and we have added the ¹H, ¹³C, ¹¹B and HMRS spectrum of **2ah** in the supporting information. Please see our revised supporting information.

Following the **procedure A** on 5 mmol scale, colorless oil liquid, yield: 46% (607.2 mg), $R_f = 0.5$ (silica gel, PE: EtOAc = 20:1, v/v), column chromatography (silica gel, PE: EtOAc = 50:1, v/v).

$^1\text{H NMR}$ (500 MHz, CDCl_3) δ 2.31 (dtd, $J = 8.0, 6.5, 1.9$ Hz, 1H), 2.22 – 2.13 (m, 1H), 1.99 – 1.91 (m, 1H), 1.88 (dd, $J = 10.7, 4.4$ Hz, 1H), 1.81 (tdd, $J = 13.3, 6.6, 2.6$ Hz, 2H), 1.48 (ddd, $J = 15.1, 11.1, 6.2$ Hz, 1H), 1.25 (s, 1H), 1.21 (s, 6H), 1.20 (s, 6H), 1.17 (s, 3H), 1.06 (s, 3H), 1.05 (s, 3H).

$^{13}\text{C NMR}$ (101 MHz, CDCl_3) δ 82.89 (s), 48.66 (s), 40.78 (s), 33.57 (s), 28.43 (s), 26.85 (s), 25.01 (s), 24.63 (s), 24.56 (s), 23.67 (s).

$^{11}\text{B NMR}$ (128 MHz, CDCl_3) δ 35.00 (s).

HRMS (EI-QTOF) m/z : $[\text{M}]^+$ Calcd. for $\text{C}_{16}\text{H}_{29}\text{BO}_2$ 264.2261; Found 264.2258.

25) Section 7.5: replace “dropwisely” with “dropwise”

Response: We sincerely thank this reviewer for pointing it out, we have corrected these mistakes. Please see our revised supporting information.

26) ^{13}C -NMR spectrum of **4ad** seems to be oddly cut. Please provide the original FID file

Response: Sorry for the negligence. We have corrected this mistake. Please see our revised supporting information.

27) Caption of NMR spectra: o-, m-, and p- should be formatted as italic

Response: We sincerely thank this reviewer for pointing it out, we have corrected these mistakes. Please see our revised manuscript as well as supporting information.

Reviewer #2 (Remarks to the Author):

The manuscript reported by Song et al. describes photo-induced trifunctionalization of bromostyrenes via formal remote radical migration reactions of tetracoordinate boron species. This is an interesting and useful piece of work on the radical chemistry in using tetracoordinate boron species. Especially, the generation of α -boryl carbon radical from gem-diboron is very important in synthetic organic chemistry. This paper is acceptable as an article of Nature communications after the following revisions.

Response: We sincerely thank this reviewer for his/her favorable comments on our submitted manuscript, we really appreciate it.

1) The title is not correct. This reaction is obviously the radical transfer reaction. So, the author should revise the title.

Response: We thank this reviewer for the constructive comments. We have corrected the title "**Photo-induced Trifunctionalization of Bromostyrenes via Remonte**

Radical Migration Reactions of Tetracoordinate Boron Species”.

2) If possible, the author should indicate the oxidation potential of borate anion **3** and the reduction potential of benzylic radical IV. In addition, although the excited-state redox potential of 4CzIPN is well known, the author should indicate its value in the manuscript, as it would be very useful for the readers.

Response: We deeply thank this reviewer for the constructive comments. We have tested the oxidation potential of complex **3**. But for the reduction potential of benzylic radical IV, we have no way to test it under the existing experimental conditions.

Cyclic voltammetry (CV) experiments were conducted in a 10 mL glass vial fitted with a glassy carbon working electrode (3 mm in diameter), a platinum wire auxiliary electrode and submerged in saturated aqueous KCl solution Ag/AgCl reference electrode. MeCN (5 mL) containing 0.02 mmol ⁿBu₄NBF₄, 0.05 mmol **3**, which was pre-prepared, were poured into the electrochemical cell. The oxidation potential of complex **3** was 1.09 V versus Ag/AgCl in MeCN.

Per the request, the oxidation potential of borate anion **3** and the excited-state redox potential of 4CzIPN were indicated in the manuscript (in the description of proposed mechanism). Please see our revised manuscript.

3) In figure 8, the structure of II is not correct. The author should revise the correct structure.

Response: Sorry for the negligence. We have corrected this mistake. Please see our revised manuscript.

4) In the scale-up experiments, the yields of **4p** and **6a** are lower than those in small-scale experiments. What are the reasons for them?

Response: We sincerely thank this reviewer for pointing it out. When we tried to

scale up the experiment, we observed that compared with small scale reactions, more *o*-boron styrene by-products were generated in the scale-up ones, resulting in a decrease in the overall yield. We tried some methods such as extraction of THF solvent, but did not significantly improve the yield. The other possibility might result from the interference of water or air during the experimental process.

5) In SI, ^{11}B NMR spectra, for example 6'a, 6'b, 6'c, 6'd, 6'e, and 6'g, are strange. Please re-check those ^{11}B NMR spectra.

Response: Sorry for the negligence. We have corrected these mistakes. Please see our revised supporting information.

Reviewer #3 (Remarks to the Author):

First of all, I would say that Song and coworkers presented a nice design of photoinduced radical transformation of tetracoordinated pinacol boronates. Different from the few previous reports, the smart utilization of a vinyl group containing aryllithiums (in-situ from aryl bromides) as the “activation reagent” allows the re-assembly of the “side” arylboronates with the generated alkyl radical to achieve a formal remote migrative bond-forming process. A very broad range of alkyl boronates were well used in this transformation. The site diversity on the olefinic aryllithium was also achievable. Gem-diborons and 1,2-diborons result in various aryl-B & alkyl-B containing diboron compounds. I believe this will be a valuable organoboron chemistry in chemical synthesis. I recommend its publication in Nature Communications after addressing the following issues.

Response: We sincerely thank this reviewer for his/her favorable comments on our submitted manuscript, we really appreciate it.

1. In Table 1, as the data are the variations from the standard conditions, please present the standard condition in the chemical equation in details. In the entries (2, 3, 4, 5), illustrating the conditions clearly about instead of what conditions of the standard.

Response: We thank this reviewer for pointing it out, we have revised some sentences and carefully checked related mistakes. Please see our revised manuscript.

2. In the proposed mechanism, the intermediate II was wrong, please modify it. The arrows between II and III should be a plus sign.

Response: We deeply thank this reviewer for pointing it out. We have corrected these mistakes. Please see our revised supporting information.

3. What about a none-terminal alkenyl group-containing aryllithium? Please present one example to show the possibility.

Response: We thank this reviewer for the constructive comments. Per the request, we used the following *o*-Br substituted styrene as the substrate, and no desired product was obtained under the standard conditions.

4. Many of the citations are not in proper format. Please check them carefully.

Response: We sincerely thank this reviewer for pointing it out, we have corrected these mistakes. Please see our revised manuscript.

REVIEWERS' COMMENTS

Reviewer #1 (Remarks to the Author):

The manuscript was carefully revised according to reviewers' suggestions. I recommend the publication in Nature Communications as it is. Congratulations to the authors for the excellent work!

Reviewer #2 (Remarks to the Author):

The author has clearly answered all the comments of the reviewers. As the result, this manuscript has become a suitable paper for Nature Communications. Therefore, this reviewer recommends its publication as a communication in Nature Communications.

Reviewer #3 (Remarks to the Author):

The concerns raised by referees have been properly addressed in this revision. Publication is recommended.